# Non-Surgical Therapy and Oral Microbiota Features in Peri-Implant Complications: A Brief Narrative Review

**DOI:** 10.3390/healthcare11050652

**Published:** 2023-02-23

**Authors:** Massimo Corsalini, Monica Montagnani, Ioannis Alexandros Charitos, Lucrezia Bottalico, Giuseppe Barile, Luigi Santacroce

**Affiliations:** 1Interdisciplinary Department of Medicine, University of Bari “Aldo Moro”, Policlinico University Hospital of Bari, p.zza G. Cesare 11, 70124 Bari, Italy; 2Department of Precision and Regenerative Medicine and Ionian Area, Section of Pharmacology, School of Medicine, University of Bari “Aldo Moro”, Policlinico University Hospital of Bari, p.zza G. Cesare 11, 70124 Bari, Italy; 3Emergency/Urgent Department, National Poisoning Center, Riuniti University Hospital of Foggia, 71122 Foggia, Italy; 4Interdepartmental Research Center for Pre-Latin, Latin and Oriental Rights and Culture Studies (CEDICLO), University of Bari “A. Moro”, 70124 Bari, Italy

**Keywords:** infection, immune response, periodontitis, peri-implantitis, antibiotics, oral microbiota

## Abstract

The therapeutic discretion in cases of peri-implantitis should take into account the limits and advantages of specific therapeutic itineraries tailored according to each clinical case and each individual patient. This type of oral pathology emphasizes the complex classification and diagnostic issues coupled with the need for targeted treatments, in light of the oral peri-implant microbiota changes. This review highlights the current indications for the non-surgical treatment of peri-implantitis, describing the specific therapeutic efficacy of different approaches and discussing the more appropriate application of single non-invasive therapies The non-surgical treatment choice with antiseptics or antibiotics (single or combined, local, or systemic) for short courses should be considered on a case-by-case basis to minimize the incidence of side effects and concomitantly avoid disease progression.

## 1. Introduction

The term “peri-implant diseases” refers to the inflammatory reactions affecting the peri-implant soft and hard tissues, including two distinct nosologically entities, peri-implant mucositis and peri-implantitis. In 2008, mucositis was defined as inflammation of the peri-implant mucosa without loss of supporting bone tissue, and peri-implantitis as a condition characterized by inflammation of the peri-implant mucosa associated with loss of supporting bone tissue. Definitions were confirmed in the consensus reports of the Sixth and of the Seventh European Workshop of Periodontology [1,2]. The reversible inflammatory reaction of the soft tissue surrounding an implant has been termed peri-implant mucositis. Instead, the term peri-implantitis refers to inflammatory processes accompanied by loss of supporting bone [3]. Detection of peri-implant mucositis is carried out with a periodontal bur in the peri-implant fissure to detect bleeding or pyorrhea. Peri-implantitis defines an inflammation of the peri-implant tissues, such as mucositis, complicated by a radiographically detectable loss of the supporting bone. Its detection is performed by measuring the drilling depth of the peri-implant fissure and attachment loss [4]. During the measurement, the existence of bleeding or pyorrhea is also evaluated. Infectious complications are one of the main causes of failure in implantology (Figure 1) [5,6,7].

Peri-implantitis in the last two decades seems to have become a major disease. The prevalence of this disease is unclear in the literature, and a recent systematic review and meta-analysis found that, on average, 19.5% of patients and 12.5% of implants had peri-implantitis. This evidence suggests that there is a high frequency of patients and implants that may be affected by this clinical condition. Hence, it suggests that booster and hygiene regimens need to be strengthened (particularly for patients with a clinical history of this disease) [8]. Infections may originate from the local bacterial biofilm and then spread in tissues surrounding osseointegration implants, causing a reversible inflammatory reaction of the peri-implant soft tissues [9]. During implant insertion procedures, mucositis manifests with a prevalence of 50%, and peri-implantitis of up to 40%. It should be considered that the implant does not behave like a totally bioinert material but triggers an immune reaction from the host [10]. In fact, at the bone–implant interface, the presence of multinucleated giant cells and macrophages, which are pathognomonic of a foreign body reaction, is a common histological observation. Therefore, osseointegration would be based on the principle of foreign body equilibrium, making it essential in the treatment of peri-implantitis to eliminate the bacterial infection in order to arrest the progression of the lesion and, to whichever possible extent, to regenerate/re-osseointegrate the peri-implant bone lost [6]. Indeed, antimicrobial therapies are often used in dentistry, both in therapeutic and in preventative regimens. Their therapeutic use focuses on the treatment of foci of inflammation in the hard or soft tissues of the oral cavity and most often are of endodontic/periodontal ethology [11]. The preventative use of antimicrobics is to protect against the effects of dental procedures and subsequent triggered infections. Indeed, microbial infections are the main cause of peri-implantitis [12], and the topical preventative antiseptic and antibiotic therapy of preference should consist of formulations allowing a slow release and longer protection. This is of particular importance when considering that microbial infection in patients with dental implants may not only initiate local peri-implantitis but also spread to distant tissues where they can induce pathological conditions (such as endocarditis) and, potentially, evolve to a systemic clinical sepsis status (Figure 2) [13,14].

Special attention should be devoted to patients with immune-compromising conditions or neoplastic diseases such as leukemias and lymphoma, as well as in subjects undergoing radiotherapy to the head and neck. Early targeted antibiotic treatment can help to avoid peri-implant bone surgery, either of the resective, conservative, or regenerative type, with the latter becoming unavoidable when the bone loss is mainly horizontal and the lost tissue cannot regenerate. The following paragraphs deal with the main non-surgical treatments to date, with particular attention to the antibiotic use and efficacy for this oral pathology [16,17].

## 2. Pathogenesis of Peri-Implant Diseases

Retrograde peri-implantitis occurs when the implant can fail due to overload, trauma, or occlusal factors. This overload occurs for four reasons, i.e., when (i) the bone in which it is located is of poor quality or of inadequate quantity, (ii) the implant is in a position such that the load is directed off axis (imbalanced distribution of occlusal forces) on the surface of the implant itself, (iii) the total number of implants is insufficient compared to the masticatory surface offered by the prosthetic superstructure, and (iv) the superstructure itself does not fit perfectly with the implants themselves [17]. These reasons will lead to periapical bone resorption without the appearance of inflammation of the peri-implant soft tissues, and therefore it will lead to a clinical picture of retrograde peri-implantitis. It differs from infectious peri-implantitis due to its association with intrasulcular microbiota more consistently with gingival health status (which may be predominantly *Streptococcus* spp.). Infectious peri-implantitis is a manifestation of periodontitis at the implant level [18,19]. These, in fact, are two diseases linked by the same clinical characteristics and the same etiological factor: the bacterial plaque. Implants have a less effective natural tissue barrier than natural teeth and are therefore less resistant to infection. This occurs for two reasons on the implant: (i) there is no cement and therefore there are also no collagen fiber insertions that run parallel and not perpendicular to the implant, and thus the only barrier that prevents the dissemination of microbes in the peri-implant sulcus is made up of circular fibers, and (ii) no biological seal (i.e., adhesion of epithelial cells via basement membrane and hemidesmosomes) develops between the soft tissues and the metallic implant surface so that the adaptation of the soft tissues to the implant surfaces would be more linked to the tone and proximity of the gingiva than to the presence of junctional epithelium attachment [20,21,22]. Therefore, for these two reasons, microorganisms find an easier way to reach the implant surface directly. Subsequently, on its implant surface, the bacteria produce endotoxins capable of initiating an acute inflammatory response, which tends to progress more apically, involving the peri-implant alveolar bone more rapidly in the destructive process. Therefore, the reduced quantitative fibroblasts/collagen ratio and the scarce blood supply of this tissue causes a lower resistance of the peri-implant supraveolar connective tissue to progressive tissue destruction. Thus, it becomes obvious that before starting implant therapy, a patient with periodontitis must be treated until its remission [18,21].

Thus, bacterial colonization of oral prosthetic and implant surfaces occurs through the formation of biofilms. Then, a stable aggregation of several microorganisms organized in a polysaccharide matrix develops to form a film adhering to a solid substrate. The biofilm represents an advantage for microorganisms (bacteria, etc.), making them more resistant to host defense mechanisms (such as phagocytes) but also to chemotherapy [23,24]. The presence of biofilm at the level of the peri-implant sulcus is to be considered a physiological phenomenon, such as the biofilm present in the gingival sulcus of the teeth. For this reason, it is important to consider the differences between the microbiota residing in the peri-implant sulcus of a healthy implant and one affected by peri-implantitis as well as the factors responsible for the transformation of a physiological biofilm into a pathogenic biofilm [25].

The oral microbiota (the largest composition of microorganisms, second only to the gut microbiota) is composed of over 700 different species of microbes, only half of which can be cultured by existing methods [26]. Potentially pathogenic bacteria found in the oral cavity include *Staphylococcus aureus*, *Streptococcus pyogenes, Streptococcus pneumoniae, Neisseria meningitidis*, *Haemophilus influenzae*, *Enterococcus faecalis*, *Enterobacteriaceae*, and *Actinomycetota* phylum [27]. The advancement of microbiological analysis techniques (following the introduction of next-generation sequencing (NGS)) has allowed more accurate analysis, overcoming the limits related to microbial culturing of species under investigation. These techniques have made it possible to identify specific periodontal pathogenic bacteria, mainly non-saccharolytic Gram-negative bacteria, associated with peri-implantitis. The NGS techniques demonstrated that microbiota in the peri-implant is different and less complex than the periodontal one, both in terms of health and disease [28]. Anaerobic Gram-negative bacteria have been identified in the peri-implant fissures of healthy implants, thus suggesting that the periodontal and peri-implant fissure/pocket constitute different niches and ecosystems. However, even a simple microbiological test can have diagnostic importance, being functional to clinical observation and radiological examination. Thus, laboratory microbial analysis of the saliva (easy to collect) and peri-implant fluid may provide useful information on the levels of different inflammatory mediators secreted in the peri-implant fluid, especially when combined with clinical signs on the state of health or pathology of the peri-implant tissue and associated oral microbiota dysbiosis [29]. The most sensitive biochemical markers include pro-inflammatory mediators such as interleukins-1β and -6, prostaglandin E2, aspartate aminotransferase, β-glucuronidase, elastase, myeloperoxidase, collagenase, and other metalloproteinases; glycosaminoglycans; and growth factors (such as TGF-β, PDGF, and TNF-α), among others [30,31]. Contamination of a transgingival implant and/or prosthetic components occurs early in what has been defined as a “race to the surface” and is influenced by the presence or absence of teeth and periodontal pockets. Interaction between microorganisms and the surface does not in fact occur through direct contact but is always mediated by salivary glycoproteins. Thus, the first stage of biofilm formation is the adsorption of salivary glycoproteins on the implant surface [32]. Protein adsorption is mainly influenced by various protein–surface interaction forces (such as Van der Waals, electrostatic, and hydrophobic forces). These proteins form a glycoprotein film on the surface that is capable of mediating the colonization of microorganisms. It contains lysozyme, histatin, staterin, amylase α, cystatin, secretory IgA, lactoferrin, and proline-rich proteins [33]. Film proteins contain various amino acid sequences capable of binding adhesion molecules expressed by early colonizers (such as bacteria). The interactions between microorganisms and acquired film are initially due to weak interaction forces (Van der Waals forces), which are subsequently replaced by irreversible chemical bonds between bacterial adhesions and receptors of the acquired film. It has been demonstrated that the acquired pellicle proteins also play a role in the metabolism of the first bacteria that colonize the implant surface (such as *Streptococcus oralis* and *Streptococcus mitis*) [34]. The metabolism of the first colonizing bacteria creates nutritional and environmental conditions more suitable for the subsequent colonizers, i.e., the late ones. Oxygen is gradually consumed by facultative aerobes and anaerobes favoring the development of late colonizers belonging to obligate anaerobic species such as *Fusobacterium nucleatum*, *Tannerella forsythensis*, *Porphiromonas gingivalis*, and *Aggregatibacter actinomycete* comitans [35]. Therefore, the late colonizing microorganisms bind to the polysaccharide chains of the biofilm or to receptors present on the capsule of the pioneer bacteria. Subsequently, thanks to coaggregation mechanisms, the formation of horizontal and vertical stratifications begins, which are suitable for the growth and multiplication of different bacterial species [36]. The result is the formation of a mature polymicrobial biofilm in which different species cohabit synergistically or antagonistically in a nutrient-rich polysaccharide matrix. The peri-implant bacteria most detected in a non-pathological condition are Gram-positive Cocci belonging to the group of *Streptococci* and *Actinomyces*, with a small presence of periodontopathogenic bacteria such as *P. gingivalis*, *T. forsythia*, *P. intermedia, Aggregatibacter*, and *F. nucleatum* [37,38].

Under healthy conditions, the resident bacteria coexist with the host, and the onset of a pathology occurs only when exogenous or endogenous factors alter this equilibrium. Some studies have shown that contamination of the surface of the implant body can occur through direct contact of the fixture with soft tissues and/or oral fluids [38]. Early implant contamination is particularly favored by contact with saliva, containing numerous adhesion proteins that favor the formation of an adhesion biofilm for bacteria (acquired biofilm) on the implant surface. However, it is unclear as to whether such initial contamination could be responsible for the early failure of an implant. Thus, a few hours after surgical placement, a layer of glycoproteins develops on the implant surface, subsequently colonized by microorganisms. During the 3-week period of plaque accumulation, the periodontal and peri-implant soft tissues develop an inflammatory infiltrate similar in composition, volume, and distance from the bone tissue. If the period of plaque accumulation is longer (3 months), the peri-implant inflammatory infiltrate, composed by plasma cells, lymphocytes, and other immune cells, is greater than the periodontal one both apically and laterally to the junctional epithelium [39]. Numerous surveys underline a great similarity between the microorganisms found in peri-implant and in periodontal lesions [40]. However, the local oral microbiota of peri-implantitis differs from that of periodontitis, whose composition may also depend on the patient’s systemic diseases [41]. Among the microorganisms found are *Peptostreptococcus micros*, *Fusobacterium Nucleatum*, *Prevotella intermedia*, and *Wolinella recta*. Changes in the oral microbiota around the implant shift the composition of local microorganisms leading to the predominance of Gram-anaerobic microbes [42]. *Aggregatibacter actinomycetemcomitans*, *Tannerela Forsythia*, *Treponema Denticola,* and also *Staphylococcus aureus* and *Enterococci* spp. are frequently isolated from peri-implant lesions [43,44,45], together with *Campylobacter rectus*, *Eikenella corrodens*, strains of the genus *Capnocytophaga*, and fungi of the genus *Candida,* all detectable in peri-implant pockets [46,47]. Interestingly, trains of the genus *Staphylococcus* and *Pseudomonas* are found not only around dental implants but also on the surfaces of other implant materials, such as those used in hip arthroplasty. In a prospective study, implants were placed and monitored weekly for four months with microbiological investigation control from the peri-implant fissure to identify the developmental diversity of the local oral microbial population [47,48]. The results showed an increased number of anaerobic microbes as early as the second week. At third week, an increase in the number of rod-shaped bacteria was observed with a corresponding decrease in Cocci bacteria. After six weeks, microbes of the *Fusobacterium* genus were detected, while at four months, *Spirochetes* spp. were also present in the peri-implant gap, with clinical signs of a 6 mm cyst and purulent exudate [49]. In another study, comparing healthy and successful implants with respect to failed implants (with pocket depths > 6 mm and bone reduction), it was found that healthy implants had a small concentration of microbes and a minor presence of rod-shaped microbes, while implants that led to failure were characterized by a high percentage of Gram-microbes, with a strong presence of *Spirochaetes* spp. and rod-shaped bacteria (Figure 3) [47,50].

Since microbial infection is a primary cause of peri-implantitis, oral hygiene is an absolute requirement. When oral hygiene is interrupted for a few days, the undisturbed accumulation of microbial plaque around implants leads to inflammation of the surrounding soft tissue. This process can be facilitated by certain risk factors and comorbidities [51,52]. It has been observed that the association of periodontitis with other risk factors, such as diabetes mellitus and smoking, increases the risk of peri-implantitis and implant loss [53]. Moreover, the genetic polymorphism linked to allele 2 of the IL1-RN gene, resulting in a greater production of IL1 during the inflammatory response of the subject (IL1 positive genotype), has been proposed as a susceptibility condition for the development of peri-implantitis. In retrospective studies, a greater loss of peri-implant bone has been observed in subjects with IL1-positive genotype and associated smoking (Figure 4) [54]. With the removal of the microorganisms (bacteria, fungi), inflammation subsides, and tissue health is restored to the area. In the presence of ligatures maintained for 6 weeks, and even after ligature removal, soft tissue inflammation and bone destruction are more marked at the implant sites. With respect to periodontal sites, where infiltrates are never in contact with the alveolar bone, the peri-implant infiltrate extends to the bone tissue (osteitis), and the osteoclasts—an expression of greater bone resorption—are more represented. Maintaining bone-level stability around an implant is a necessary requirement for successful post-implant repair [55].

## 3. Diagnostic Evidence of Peri-Implantitis

As briefly anticipated, the synergy between microbiological results and early detection of clinical signs can be of fundamental importance for the diagnosis and the prognostic outcome of peri-implantitis [58]. At present, a unifying criterion for the diagnosis of peri-implantitis based on the calculation of the loss on the peri-implant bone support is not considered feasible, since the bone remodeling that leads to the resorption of the cervical bone of the implant in the most apical area is a pathophysiological event naturally occurring after an implant placement. For this reason, each proposed diagnostic method is based on specific criteria [59]. Vertical bone loss of less than 1.5 mm in the first year after implant placement and less than 0.2 mm in each subsequent year was one of the first criteria used to characterize an implant as successful [60]. More recently, an attempt has been made to classify peri-implantitis in the context of epidemiological studies investigating risk factors and causes of implant failure (Figure 4) [61].

The way in which to measure the depth of the peri-implant fissure or pocket is also determinant in the diagnosis of peri-implant disease mucositis or peri-implantitis. In peri-implantitis, the severity of the disease and its classification as an initial or advanced form leads to different prognostic conclusions [62]. As an example, during measurement within follicles with peri-implant disease, the tip of the molar ends very closely to the bone edge (less than 0.5 mm away), while in healthy peri-implant spaces, the distance measured from the bone is in between 0.5 and 1.5 mm [63,64]. In several studies, the presence of peri-implantitis was considered only in cases with radiographically detected vertical loss of peri-implant bone, measured from the fixed points of the implants. The more correct diagnosis, however, would come from radiographical comparison of the implant at two subsequent time intervals in which the loss of the peri-implant bone is evident, bearing in mind that a certain absorption of the tissue bone should be considered normal [65]. Clinically, it is possible to detect soft tissue subsidence around an implant, whose presence in the anterior area can create an aesthetic problem [66]. In a study conducted on the correlation of the depth of the peri-implant pocket with the peri-implant bone loss, measurements were performed with and without prosthetic restoration. The level of bone around the implants can be significantly correlated with the existence of keratinized mucosa and with the appearance of pyorrhea from the peri-implant fissure. The absence of bleeding on palpation with a periodontal probe has significant prognostic value on the healthy state of periodontal tissues [67,68]. Conversely, the presence of bleeding, especially if combined with positive microbiological assessment, can be an indication of disease progression. Finally, in another study, a more detailed evaluation, according to bone morphology damage around the implant, was added to the assessment of vertical loss tissues. This additional categorization describes the damage at two levels, horizontal (category I) and vertical (category II). The horizontal category I can have three further sub-categories when the buccal bone wall (a) is absent without a circular peri-implant bone loss (class Ib), (b) is absent with a circular peri-implant bone loss (class Ic), and (c) when loss occurs around the implant (class Ie) [68,69,70,71]. Similar to this, another categorization of peri-implantitis classifies the type of bone damage surrounding implants as follows: (a) circular crater bone lesion with four bony walls; (b) circular crater bone lesion with three bony walls; (c) circular crater bone lesion with two bony walls; (d) circular lesion resembling a bone crater with a bony wall; (e) bone damage with cleft defect on an implant surface; and (f) in case there is no bone wall around the implant, bone loss is characterized as horizontal [70,71,72].

## 4. The Non-Surgical Peri-Implantitis Treatment

In general, on the basis of the evaluation of the fundamental diagnostic parameters for peri-implantitis, it is possible to proceed with the adoption of the CIST (Cumulative Interceptive Supportive Therapy) therapeutic protocol to prevent and/or block the development of peri-implant lesions. It is a cumulative protocol because it consists of five specific pathways associated in sequence with an increasing antibacterial potential proportional to the severity and extent of the lesion: mechanical detoxification (A), antiseptic therapy (B), antibiotic therapy (C), surgical therapy (D), and explant (E) [73].

### 4.1. Mechanical Debridement

The treatment of peri-implantitis (the same as periodontitis) is divided into two main categories (non-surgical and surgical treatment), which further include various therapeutic techniques. The non-surgical approach to peri-implantitis consists of a synergy of procedures summarized in Figure 5 [74]. Mechanical debridement aims to remove microbes, soft and hard chemical deposits from the implant surfaces, and smooth/polish these surfaces mechanically. The removal of the microorganisms favors the inflammation subsiding, allowing tissue health restoration and ameliorating the eradicating effects of local or systemic antibiotic treatment [75,76].

Curettes are the basic treatment for the removal of tartar and plaque deposits from the implant surface, being performed manually with instruments made of different materials. Titanium-coated curettes have similar hardness to the implant surface and therefore are not dangerous for the implant itself. Carbon fiber curettes are used to remove bacterial deposits without damaging the implant surface. Teflon curettes share the same characteristics and are often proposed in combination with air/polishing systems. Plastic curettes have the most limited debriding capacity [76].

Alternative methods employ equipment based on concepts of vibrational frequencies in which the working part oscillates at sonic or ultrasonic frequencies. The sonic ones are instruments provided with a compressed air handpiece, wherein the air pressure mechanically creates vibrations (2500–6300 cycles/s, 6–8 kHz) or oscillation (50–200). The ultrasonic instruments are characterized by different vibration parameters (25,000–42,000 cycles/s, 25–42 kHz). At these frequencies, heat is produced, and therefore cooling is required. They may have a direct or indirect (cavitation) action and a cleansing effect [77].

### 4.2. Non-Ionizing Radiation Sources

The laser is a device emitting a coherent, monochromatic, and concentrated straight beam of light that is extremely collimated. In a bactericidal modality, CO_2_, diode, erbium-doped yttrium aluminum garnet laser (Er: YAG), and erbium chromium-doped yttrium scandium gallium garnet (Er, Cr: YSGG) lasers are used in the treatment of peri-implant diseases with increasing frequency. Compared to mechanical debridement with the plastic curettes, the use of Er: Yag has led to significantly better results in terms of bleeding index. Although there are some data in contrast, laser therapy should be considered as an adjuvant rather than a treatment option. Further studies are needed to evaluate the clinical efficacy of laser therapy as a treatment for peri-implantitis [77,78].

Photodynamic therapy (PDT) generates reactive oxygen species with the aid of high-energy single frequency light, for example, a laser diode (wave 580–1400 nm), in combination with photosensitizers (such as toluidine blue concentration between 10 and 50 µg/mL). This type of treatment generates bactericidal effects against aerobic and anaerobic bacteria (such as *Aggregatibacter Actinomycetemcomitans, P. gingivalis*, *P. intermedia*, *S. mutans*, and *Enterococcus Faecalis*) [79].

### 4.3. Antiseptics

Slow-release antibacterial agents include chlorhexidine, indicated for the adjunctive topical and antimicrobial treatment of both moderate-to-severe adult periodontitis and peri-implantitis [80]. Chlorhexidine can be used as a mouthwash at 0.12% as a bacteriostatic and at 0.2% as a bactericide. Chlorhexidine gel xanthan is a product indicated for the topical treatment of periodontitis and peri-implantitis. The gel may have 1.5% or 0.5% chlorhexidine in the rapid-release digluclonate form (high concentration in the first 24 h and continuous release for approximately fifteen more days), and 1% in the slow-release dichlorhydric form. The use of this antiseptic agent (0.1–0.2% chlorhexidine digluconate) is indicated for a period of 3–4 weeks, with additional local lavage with 0.1–0.5% chlorhexidine solution or local application of 1% chlorhexidine gel, and should be accompanied by the simultaneous repetition of the oral hygiene instructions [81,82]. While for shallow peri-implant pockets (<4 mm), the sole use of mechanical techniques might be sufficient, the combined application of a 0.2% chlorhexidine solution may provide advantages in the treatment of deep (>5 mm) peri-implant pockets. Thus, chlorhexidine can be useful in all peri-implant pockets [83] and also when the patient presents an epi-implant plaque, bleeding on detection, possible effusion (unnecessary), deep peri-implant pockets (>5 mm), and radiographically visible peri-implant bone loss. The final step required, after cleaning the implant surface area with mechanical debridement and the use of antiseptics, should be the use of antibiotics (with local application or systemic administration).

### 4.4. Antibiotics

Antibiotics are mostly derived from living organisms such as bacteria and fungi, while only a few are of synthetic or semi-synthetic origin. These drugs are used to enhance the antibacterial effect of mechanical debridement and to prevent bacterial recolonization of the implant surface [84]. Antibiotics can be administered topically (gel or microspheres with slow release) or systemically (oral or injective routes). Topical antibiotic administration allows for the selection of the sites to be treated by prolonging contact with the pathogen and to maintain a concentration adequate and constant over time, whereas systemic administration should be performed to reach sites that would normally be inaccessible [85]. As emerged from the systematic reviews to date, no randomized controlled clinical trials have been carried out reporting the systemic administration of antibiotics as adjunctive therapeutic agents to the use of mechanical or antiseptic/local antibiotics for the non-surgical treatment of peri-implantitis [86]. As for the topical antiseptic therapy, the antibiotic treatment indicated more in peri-implantitis should be characterized by a slow-release formulation to maintain the efficacy in the pocket [87].

Metronidazole is a derivative of 5-nitroimidazole. When the nitro group is reduced to reactive radical species, these radicals react with cellular components such as DNA or protein. On the basis of metronidazole’s mechanism of action, the metronidazole-based film products are especially active on Gram-negative anaerobic bacteria. These products consist of an absorbable gel containing 25% metronidazole benzoate in a matrix made of a mixture of glyceryl-mono-oleate and oil of sesame [88,89]. The decay of the antibiotic (release is effective over a time frame ranging from 24 to 36 h) and the concentration in the peri-implant pocket follow an exponential trend. The application can be repeated once a week. Several studies have reported additional benefits from combination with topical slow-release doxycycline or the placement of tetracycline fibers within peri-implant pockets.

Tetracyclines’ antimicrobial activity results from drug binding to the 30S subunit of the ribosome in susceptible bacteria, with subsequent interference with bacterial protein synthesis. Tetracycline preparations contain broad-spectrum antibiotics effective against anaerobic organisms and facultative anaerobes with a high capacity to bind to the dentin, favoring maintenance of the antibacterial action over time [88]. Generally, the active principle tetracycline is contained in cellulose acetate fibers, polyacrylic acid strips, collagen preparations, and hydroxypropyl cellulose films with poly(methacrylic acid). The most used system is based on cellulose fibers loaded with tetracycline hydrochloride, to be inserted into the pocket. One topical application of tetracyclines consists of fibers of ethylene-vinyl-acetate containing 25% (weight/volume) of tetracycline hydrochloride. The fibers are wrapped around the implant in several concentric layers until the peri-implant space is totally filled; once the placement is completed, an isobutyl-cyanoacrylate adhesive is applied to the mucosal margin to fix the fibers. In case of fiber loss during the seven days following placement, other fibers can be applied. The fibers are usually removed after 10 days [89,90,91,92,93].

Doxycycline, another tetracycline, is applied directly in the form of gel to infected sites. This formulation provides very high antibiotic concentrations with a duration of up to seven days. Typically, the application kit consists of two syringes, one containing the polymer liquid, the other the antibiotic powder. Treatment with chlorhexidine gel or mechanical therapy and subsequent slow-release doxycycline polymer improves healing more significantly than mechanical therapy alone [93].

Products based on minocycline, a broad-spectrum antibiotic of the tetracycline class, are active against Gram-negative and Gram-positive bacteria [94]. The topical application of minocycline microspheres improves the therapeutic effect more than the topical application of chlorhexidine gel in incipient or intermediate peri-implant cysts, while the additional efficacy of topical application of minocycline in deep (>5 mm) peri-implant pockets is not clear [95]. This formulation consists of granules of different sizes of polymer with active principle microencapsulated and is administered subgingivally after scaling; the release takes place in the crevicular fluid, and detectable drug levels persist for at least fourteen days and they are completely bioabsorbable. Another topical aid for the treatment of peri-implantitis is based on the treatment with minocycline in biodegradable microspheres combined with mechanical debridement, being able to improve clinical parameters at 12 months follow-up [96,97].

Penicillines inhibit the biosynthesis of the mucoproteins constituting the cell wall of susceptible bacteria by interacting with the so-called penicillin-binding proteins (PBP). Slow-release, piperacillin-based antibacterials are available as fluid-alcohol-based solutions containing 12% acrylic resins and 10% piperacillin sodium. Piperacillin has a spectrum of activity that includes Gram-positive and Gram-negative bacteria. The product solidifies rapidly after application, creating a protective film (made up of the two polymers) by evaporation of the organic solvent (ethyl alcohol). The distinctive feature of this film is to release the antibiotic in a controlled way (piperacillin sodium), with it remaining inside the resins after evaporation of the organic solvent [88,93]. These resins are permeable and insoluble to water, and this feature allows for the formation and permanence of the film in the oral cavity where it is maintained for 7–10 days with the release of piperacillin sodium.

The beneficial effects of systemic antibiotic administration (such as metronidazole 250 mg twice a day or clavulanic acid 500 mg twice a day/10 days) in peri-implantitis, with the exception of possible abscess episodes, still remains uncertain. The most used with beneficial effects of systemic antibiotic administration are those based on metronidazole (3 × 350 mg or 2 × 250 mg per day) or a combination of metronidazole (500 mg per day) and amoxicillin (375 mg per day) during the last 10 days of therapy. Another antibiotic that can be administered is ornidazole (2 × 500 mg per day) or clavulanic acid (500 mg twice a day/10 days). Therefore, when peri-implantitis is localized and is not accompanied by a widespread periodontal problem with the presence of other infected sites, the use of local antibiotics takes place. These must remain in situ (at least 7–10 days) in concentrations in order to penetrate the biofilm of the submucosa so they have good antimicrobial efficacy as, for example, occurs with tetracycline fibers (Figure 6) [98,99,100,101,102,103,104,105].

Although topical application of antibiotics may offer hypothetical additional benefits, in the non-surgical treatment of peri-implantitis, some points still need to be clarified: for example, the depth of the peri-implant pockets that should indicate whether or not topical antibiotic treatment should be used is still an unresolved question. Consequently, the synergy of microbiological and clinical signs can be useful for the diagnosis and prognosis of peri-implantitis. Therefore, before deciding to treat peri-implantitis with adjunctive antibiotic therapy, the microbiological research and analysis may provide valuable information to guide on the selection of the appropriate antibiotic drug, route of administration, and antibiotic regimen. However, the continuous emergence of antibiotic-resistant bacterial species makes it necessary to limit the use of antibiotics in periodontal therapy [98,99].

However, microbiological examination alone can be considered a diagnostic method because it serves clinical observation and radiological examination. Thus, laboratory microbial research analysis of the saliva (is easier to collect) and peri-implant fluid can investigate the possible correlation between the levels of different of biochemical markers of inflammation (inflammatory mediators) secreted in the peri-implant fluid with clinical signs that indicate the state of health of the peri-implant tissue and its pathological changes, together with the oral microbiota dysbiosis [62]. The biochemical markers are pro-inflammatory mediators (interleukins-1β and -6, prostaglandin E2), aspartate aminotransferase, β-glucuronidase, elastase, myeloperoxidase, collagenase, various other metalloproteinases, glycosaminoglycans, and growth factors (i.e., TGF-β, PDGF, TNF-α), among others [63,64].

## 5. Conclusions

For correct management of the prosthetic implant, the most important tool to apply is prevention. This starts right from the beginning of the drafting of the prosthetic implant rehabilitation plan. The correct design of the prosthesis must avoid occlusal overloads that can alter the bone–implant interface. Another factor would be the elimination or reduction of the various risk factors (for example smoking) with effective and regular professional and home oral hygiene interventions. In non-surgical therapy, the combination of mechanical cleaning techniques and air polishing systems may be recommended together in the short term with the aid of antiseptics or antibiotics (systemic or local). These procedures are effective for bacterial load reduction in combination with the other techniques. However, long-term benefit results of these techniques are still scarce. Moreover, on the basis of available data, non-surgical therapy of peri-implantitis may be not effective in resolving the disease, as only limited improvements have been reported in the main clinical parameters and there is a clear tendency towards recurrence. In part, this may depend on the stage of the disease at the time of diagnosis, or insufficient care on prevention procedures (including oral hygiene). For all these cases, it is recommended to consider advanced therapies, including surgical procedures, when non-surgical peri-implant therapy is unable to achieve significant improvements in clinical parameters.

In cases where the use of antibiotics is considered necessary, local application is the first option when the pathological condition is localized, as the systemic administration in these cases may lead to exposure to unnecessary side effects. On the other hand, more complicated and recurrent conditions may require systemic administration on the basis of the more appropriate antibiotic choice in light of the mechanism of action and individual patient’s characteristics. In this context, development of reliable biochemical assays would greatly enhance the diagnosis and follow-up of periodontitis or peri-implantitis progression. Finally, since local oral microbiota dysbiosis has been indicated as an important contributor to peri-implant inflammation, further studies should consider this additional aspect.

## Figures and Tables

**Figure 1 healthcare-11-00652-f001:**
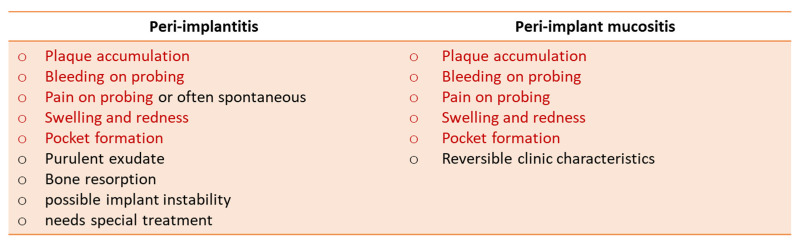
The clinical signs among peri-implantitis and peri-implant mucositis (in red are the common clinical features).

**Figure 2 healthcare-11-00652-f002:**
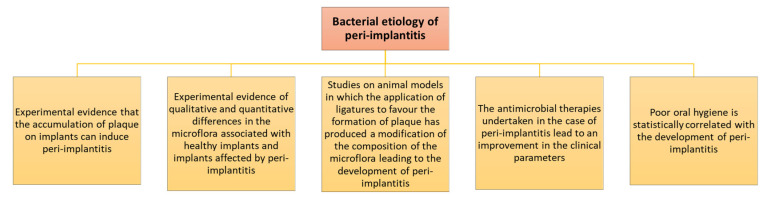
According to the most widespread and thus far most documented theory, bacterial colonization of the implant surface represents the primary etiological factor of peri-implantitis. Different microbial species, in particular, Gram-negative anaerobes, would be at the basis of the development and progression of peri-implantitis [14,15].

**Figure 3 healthcare-11-00652-f003:**
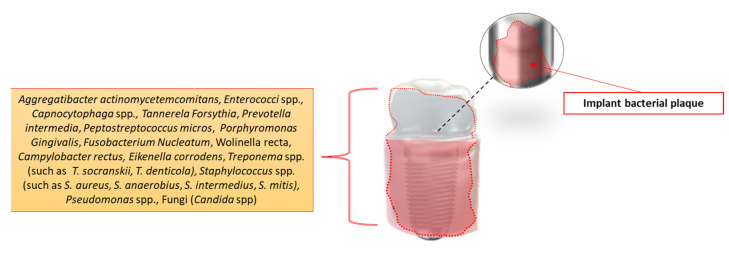
The main microorganisms found in peri-implant lesions [28,39,40,41,42,43,44,45,46,47,48,49,50].

**Figure 4 healthcare-11-00652-f004:**
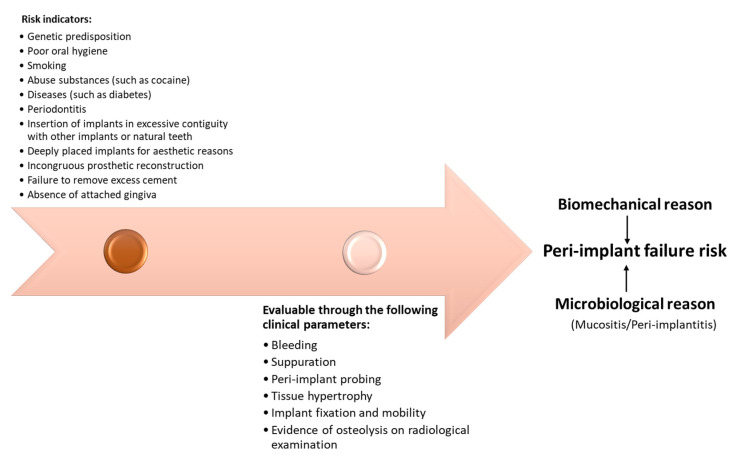
Implant failure risk can be early or late. Early implant failure, due to lack of osseointegration, may be attributable to factors not related to susceptibility to periodontitis, such as overheating of the bone > 47° during the preparation of the implant site, with consequent peri-implant bone necrosis, an early infection, a lack of primary stability, or early mechanical overload. Late implant failure, once osseointegration has taken place, can be of a biomechanical, aesthetic, or biological nature. The major contraindications relating to the implant may be insulin-dependent diabetes, osteoporosis, heart disease, pregnancy and breastfeeding, acute articular rheumatism, trigeminal neuralgia, or bruxism [53,54,55,56,57].

**Figure 5 healthcare-11-00652-f005:**
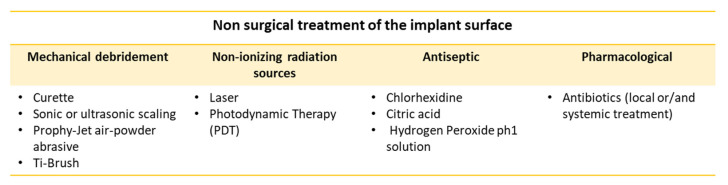
Summary of the non-surgical peri-implantitis treatments.

**Figure 6 healthcare-11-00652-f006:**
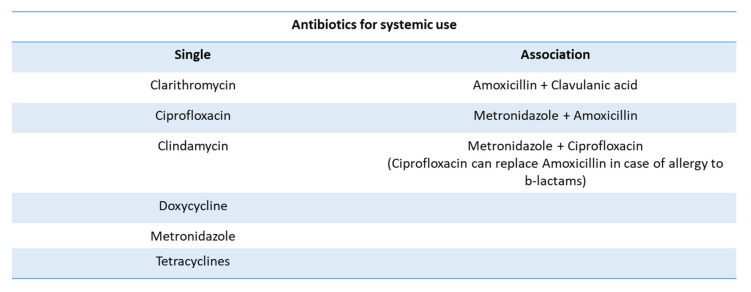
The types of antibiotics most commonly used in the case of an oral implant infection.

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
