# Peer review of "Non-Surgical Therapy and Oral Microbiota Features in Peri-Implant Complications: A Brief Narrative Review"

_healthcare, 2023, doi:10.3390/healthcare11050652_

Round 1

Author Response

The manuscript titled "Antibiotic therapy and oral microbiota features in peri-implantitis" is a concise review article focusing on antibiotic therapy for oral microbial infections associated with periimplantitis. The article is sharp and scientifically sound to the point. The article is well written. The article just needs some minor modifications to make it more interesting for the readers. Following are my suggestions to be incorporated in the manuscript,

  1. The introduction needs to be modified. The authors haven’t mentioned clearly the statement of the problem like, why antibiotic therapy is better for oral microbial etc. Try to add a few more works of literature to support the statement

First of all, thank you very much for your useful advices on our manuscript. Following your suggestions, the text has been modified and new additional references and one new figure have been included.

  1. The background of biomedical needs to be still elaborated for a better understanding of the biological conditions which need this approach.

This paragraph has been implemented, with additional references included. Changes are highlighted in green and deleted lines highlighted in red.

  1. 2-3 lines in the last paragraph stating the hypothesis of the review and what ways this article will stand alone from previously published literature on this topic.

The introduction has been rephrased and the aim of this work highlighted (in green)

  1. The pathogenesis paragraph needs to include some schematic figures or tables including the various studies on pathogenesis

Two figures have been included to describe this point. (highlighted in green)

  1. The authors can include the following references in the non-surgical section;

Polymeri A, van der Horst J, Anssari Moin D, Wismeijer D, Loos BG, Laine ML. Non‐surgical peri‐ implantitis treatment with or without systemic antibiotics: a randomized controlled clinical trial. Clinical Oral Implants Research. 2022 May;33(5):548-57.

Roccuzzo A, Klossner S, Stähli A, Imber JC, Eick S, Sculean A, Salvi GE. Non‐surgical mechanical therapy of peri‐implantitis with or without repeated adjunctive diode laser application. A 6‐month double‐blinded randomized clinical trial. Clinical oral implants research. 2022 Sep;33(9):900-12.

Done as suggested

Reviewer 2 Report

This a narrative short review about the antibiotic therapy and oral microbiota features in peri-implantitis. The manuscript should be revised by an English editing company as it includes several typos and grammatical errors. In addition, it should be reformulated for short paragraphs and to include tables and brilliant figures.

Author Response

This a narrative short review about the antibiotic therapy and oral microbiota features in peri-implantitis. The manuscript should be revised by an English editing company as it includes several typos and grammatical errors. In addition, it should be reformulated for short paragraphs and to include tables and brilliant figures.

The Authors are grateful for the Reviewer’s helpful suggestions. According to his/her requirements, an extensive revision of the English form has been performed and tables, highlighted in green, have been added.

Reviewer 3 Report

In ‘Antibiotic therapy and oral microbiota features in peri-implantitis. A short review.’ Corsalini et al. review non-surgical therapy for treatment of peri-implantitis. While the topic sounds interesting, the short review could be significantly revised to improve value to potential readers.

Specific comments:

1. The second half of the abstract should be rewritten to use complete and grammatically correct sentences.

2. The last two sentences in the introduction (about the oral microbiota and the potentially pathogenic bacteria) seem out of place and should be better linked to the introductory material.

3. In section 2, around the lines 90-92, it is not clear what exactly is the cause of peri-implant diseases.

4. The Figure 1 is not particularly informative.

5. The description of the study results in lines 139-142 is unclear.

6. In lines 114-177, relatively a lot of text is spent describing how peri-implantitis is diagnosed. This paragraph could be substantially shortened as it is tangential to the main topic of the review.

7. The section 4, which discusses the non-surgical peri-implantitis treatment, this section should be expanded and better organized, as it is the main focus of the review.

o   Several antibiotic/antimicrobial compounds are covered in this section, but it is not clear how they are organized.

o   The section covers several different methods for delivery of these compounds, but it is not always clear what is used clinically vs. what is being researched and what approaches are more promising than others.

o   Some information is repeated multiple times in slightly different ways – this could be streamlined.

o   There are some sentences on the use of microbiological testing for the diagnosis/analysis, but this could be better covered in a separate paragraph or even subsection.

8. The conclusions are not particularly supported by the rest of the review and are not particularly coherent with the title of the review. The main conclusion is that advanced and/or surgical therapies may need to be considered, but these are not really discussed in the review. Further, the conclusions are making recommendations to the clinical practitioner, but it is not clear from the rest of the review if the studies evaluated provide robust clinical data to support these conclusions.

Author Response

In ‘Antibiotic therapy and oral microbiota features in peri-implantitis. A short review.’ Corsalini et al. review non-surgical therapy for treatment of peri-implantitis. While the topic sounds interesting, the short review could be significantly revised to improve value to potential readers.

Specific comments:

  1. The second half of the abstract should be rewritten to use complete and grammatically correct sentences

The Authors agree with the Reviewer’s comments. An extensive revision of the English form has been performed on the whole manuscript and the abstract sentences rewritten (highlighted in green)

  1. The last two sentences in the introduction (about the oral microbiota and the potentially pathogenic bacteria) seem out of place and should be better linked to the introductory material

The suggested changes have been done

  1. In section 2, around the lines 90-92, it is not clear what exactly is the cause of peri-implant diseases

 An extensive revision of the English form has been performed on the whole manuscript and the specific section redrafted (highlighted in green)

  1. The Figure 1 is not particularly informative

Figure 1 has been replaced with new Figure 4

  1. The description of the study results in lines 139-142 is unclear.

Text has been rephrased

  1. In lines 114-177, relatively a lot of text is spent describing how peri-implantitis is diagnosed. This paragraph could be substantially shortened as it is tangential to the main topic of the review

Text has been rewritten and few sentences deleted (highlighted in red)

  1. The section 4, which discusses the non-surgical peri-implantitis treatment, this section should be expanded and better organized, as it is the main focus of the review.  Several antibiotic/antimicrobial compounds are covered in this section, but it is not clear how they are organized.  The section covers several different methods for delivery of these compounds, but it is not always clear what is used clinically vs. what is being researched and what approaches are more promising than others.

The whole Section 4 has been reorganized and rewritten as required, with paragraphs now including more detailed information and new references (highlighted in green, yellow, and deleted text in red)

  • Some information is repeated multiple times in slightly different ways – this could be streamlined.

An extensive revision of the manuscript has been performed and several repeated concepts eliminated (text highlighted in red)

  • There are some sentences on the use of microbiological testing for the diagnosis/analysis, but this could be better covered in a separate paragraph or even subsection

This part has been moved to section 2 (highlighted in yellow).

  1. The conclusions are not particularly supported by the rest of the review and are not particularly coherent with the title of the review. The main conclusion is that advanced and/or surgical therapies may need to be considered, but these are not really discussed in the review. Further, the conclusions are making recommendations to the clinical practitioner, but it is not clear from the rest of the review if the studies evaluated provide robust clinical data to support these conclusions.

The Authors are grateful for this comment. The main point of this review is the description of methods used on non-surgical therapy, when and if the lesion can be prevented or reduced or healed before approaching invasive methods. According to Reviewer’s suggestions, the conclusions have been modified for better clarity ( highlighted with green, yellow, and red deleted).

Reviewer 4 Report

The article presented for peer review does not present any new or groundbreaking information, moreover this reviewer performed a search on Epistemonikos, using Mesh keywords provided by the authors, which yielded many relevant studies – mostly systematic reviews covering this topic in much broader spectrum.

Therefore this Reviewer believes that presented manuscript is not fit for publication, especially in an impacted journal.

Author Response

The article presented for peer review does not present any new or groundbreaking information, moreover this reviewer performed a search on Epistemonikos, using Mesh keywords provided by the authors, which yielded many relevant studies – mostly systematic reviews covering this topic in much broader spectrum.

Therefore this Reviewer believes that presented manuscript is not fit for publication, especially in an impacted journal.

The Authors are sorry for this evaluation. However, with the extensive revision performed and with several novel information provided, we hope that our manuscript has been sufficiently improved to merit publication in this Journal.

Reviewer 5 Report

Antibiotic therapy and oral microbiota features in peri-implantitis. A short review.

Reviews in scientific research are tools that help synthesize literature on a topic of interest and describe its current state. Different types of reviews are conducted depending on the research question and the scope of the review.

For the chosen topic it is best to perform a systematic review.

The title is appropriate

It is well written

It does not have clear objectives, nor does it raise any hypothesis.

It should be structured with inclusion and exclusion criteria.

A systematic review is one such review that is robust, reproducible, and transparent. It involves collating evidence by using all of the eligible and critically appraised literature available on a certain topic. A systematic review should be reconsidered to give it more value.

Author Response

Reviews in scientific research are tools that help synthesize literature on a topic of interest and describe its current state. Different types of reviews are conducted depending on the research question and the scope of the review. For the chosen topic it is best to perform a systematic review. The title is appropriate. It is well written. It does not have clear objectives, nor does it raise any hypothesis. It should be structured with inclusion and exclusion criteria.

A systematic review is one such review that is robust, reproducible, and transparent. It involves collating evidence by using all of the eligible and critically appraised literature available on a certain topic. A systematic review should be reconsidered to give it more value.

The Authors agree on the concept that a systematic review would be more complete and informative. However, we would like to point out that our article has been structured as a narrative and short review, highlighting few but significant concepts that may be useful for practitioner and easy to read. With the extensive revision performed and with several novel information provided, we hope that our manuscript has been sufficiently improved to merit publication in this Journal.

Round 2

Reviewer 2 Report

Accept in the present form

Author Response

Dear Reviewer 2, 

thank you very much for accepting our manuscript on behalf of all the authors. 

Reviewer 3 Report

While the authors have improved the manuscript with revision, it is still not of a particularly high quality as a short review.  In particular:

(1) From the title, it appears that a major focus of the review will be on antibiotic therapy, but a significant portion of the review is still dedicated to the definition and diagnosis of peri-implantitis or peri-implant mucositis. These introductory sections could be further condensed.

(2) If peri-implantitis and peri-implant mucositis are defined as different things and both will be addressed in the review, this should be reflected in the title.

(3) The information regarding the causes of these disease and their prevalence (e.g., p. 2) should be more precise, referring to primary studies that identify these causes or determine these numbers, rather than referring to other reviews (this is a key sign that the content of the review has been covered elsewhere).

(4) In the section on pathogenesis of peri-implant diseases, care should also be taken to avoid simply providing a summary of other reviews.

(5) Further, it should be more clear what information about the pathogenesis of these diseases has been determined clinically compared to what is hypothesized, for example, from animal studies.

(6) Likewise, the section on the diagnosis of the disease is heavily focused on information from other reviews rather than from primary source material. 

Author Response

Dear Reviewer 3,

With this cover letter we address point-by-point your useful comments. In bold style you can find our modifications. Thank you for considering our work, helping us to improve this manuscript.

While the authors have improved the manuscript with revision, it is still not of a particularly high quality as a short review.  In particular:

  • From the title, it appears that a major focus of the review will be on antibiotic therapy, but a significant portion of the review is still dedicated to the definition and diagnosis of peri-implantitis or peri-implant mucositis. These introductory sections could be further condensed.

We have previously shortened it and the title of the narrative review has been changed because of this. See all current changes highlighted purple.

  • If peri-implantitis and peri-implant mucositis are defined as different things and both will be addressed in the review, this should be reflected in the title.

We changed the text because of this. See all the changes in purple.

  • The information regarding the causes of these disease and their prevalence (e.g., p. 2) should be more precise, referring to primary studies that identify these causes or determine these numbers, rather than referring to other reviews (this is a key sign that the content of the review has been covered elsewhere).

We have added additional information. See all the changes in purple colour.

  • In the section on pathogenesis of peri-implant diseases, care should also be taken to avoid simply providing a summary of other reviews.

We have added additional information. See all the changes in purple colour.

  • Further, it should be more clear what information about the pathogenesis of these diseases has been determined clinically compared to what is hypothesized, for example, from animal studies.

This is a narrative review, but we have added more additional information about human evidences. See all the changes in purple colour.

  • Likewise, the section on the diagnosis of the disease is heavily focused on information from other reviews rather than from primary source material.

We have revised the text accordingly.

Reviewer 4 Report

The Authors have made efforts in improving the paper, however in my opinion the changes are still insufficient for such manuscript to be considered for publication in a highly impacted journal.

I do understand that this is meant to be a narrative review, unfortunately it does not bring any novel information, moreover even if we take into account the possibility of the paper being read by clinicians, the information is still partial and biased. Clinicians benefit much more from clear guidelines from working groups dedicated to the use of antibiotics in certain conditions.

Author Response

Dear reviewer 4,

First of all, thank you very much for your consideration on behalf of all the authors. Here it is, in bold, the address to your comment.

The Authors have made efforts in improving the paper, however in my opinion the changes are still insufficient for such manuscript to be considered for publication in a highly impacted journal. I do understand that this is meant to be a narrative review, unfortunately it does not bring any novel information, moreover even if we take into account the possibility of the paper being read by clinicians, the information is still partial and biased. Clinicians benefit much more from clear guidelines from working groups dedicated to the use of antibiotics in certain conditions.

This is a narrative review, but we have added additional information with a new table. See all the changes in purple colour.

Reviewer 5 Report

The work has been improved, but does not meet sufficient criteria, and is a very partial view of Antibiotic therapy and oral microbiota features in peri-implantitis. Brief review.

Author Response

Dear reviewer 5, 

thank you for such consideration, here it is the address to your comment. 

The work has been improved, but does not meet sufficient criteria, and is a very partial view of Antibiotic therapy and oral microbiota features in peri-implantitis. Brief review.

We have added additional information with a new table. See all the changes in purple colour.